# Growth from the Melt and Properties Investigation of ScF$_3$ Single Crystals

**Denis Karimov** [1],*, **Irina Buchinskaya** [1], **Natalia Arkharova** [1], **Pavel Prosekov** [1],
**Vadim Grebenev** [1], **Nikolay Sorokin** [1], **Tatiana Glushkova** [2] and **Pavel Popov** [3]

[1]   Shubnikov Institute of Crystallography of Federal Scientific Research Center «Crystallography and Photonics», Russian Academy of Sciences, Leninskiy Prospekt 59, Moscow 119333, Russia
[2]   Lomonosov Moscow State University, GSP-1, Leninskie Gory, Moscow 119991, Russia
[3]   Petrovsky Bryansk State University, Bezhitskaya str. 14, Bryansk 241036, Russia
*   Correspondence: dnkarimov@gmail.com

**Abstract:** ScF$_3$ optical quality bulk crystals of the ReO$_3$ structure type (space group $Pm\bar{3}m$, $a = 4.01401(3)$ Å) have been grown from the melt by Bridgman technique, in fluorinating atmosphere for the first time. Aiming to substantially reduce vaporization losses during the growth process graphite crucibles were designed. The crystal quality, optical, mechanical, thermal and electrophysical properties were studied. Novel ScF$_3$ crystals refer to the low-refractive-index ($n_D = 1.400(1)$) optical materials with high transparency in the visible and IR spectral region up to 8.7 μm. The Vickers hardness of ScF$_3$ ($H_V \sim 2.6$ GPa) is substantially higher than that of CaF$_2$ and LaF$_3$ crystals. ScF$_3$ crystals possess unique high thermal conductivity ($k = 9.6$ Wm$^{-1}$K$^{-1}$ at 300 K) and low ionic conductivity ($\sigma = 4 \times 10^{-8}$ Scm$^{-1}$ at 673 K) cause to the structural defects in the fluorine sublattice.

**Keywords:** scandium fluoride; growth from the melt; Bridgman technique; bulk crystals; thermomiotic materials; negative thermal expansion; optical materials; crystal characterization; refractive index; thermal conductivity; ionic conductivity; microhardness

## 1. Introduction

Crystalline materials based on the rare-earth fluorides $R$F$_3$ (where $R$ = La – Lu, Y, Sc) play a significant role in the photonics and optoelectronic instrument engineering for many years [1]. ScF$_3$ is one of the most unusual and insufficiently explored fluoride. It is a thermomiotic, i.e., it features a negative thermal expansion (NTE) coefficient over a wide temperature range from 10 to 1100K [2]. The stability of the $R$F$_3$ structural type under standard conditions changes in the rare-earth row as $R^{3+}$ ionic radius decreases [1]:

$$\text{LaF}_3\text{-type (tysonite)} \rightarrow \beta\text{-YF}_3 \text{ (Fe}_3\text{C-cementite)} \rightarrow \text{ReO}_3.$$

In this case coordination number (CN) of the rare-earth cations changes, respectively, from CN = 11 (LaF$_3$-type) to CN = 8–9 (β-YF$_3$-type), and then up to CN = 6 (ReO$_3$-type). According to the crystal-chemical characteristics, ScF$_3$ differs drastically from other $R$F$_3$, and is close to InF$_3$, but its compounds have high chemical stability. In the $R$F$_3$ row, it is the only one that has structural ReO$_3$ type and it belongs to the perovskite-like compounds family with the common formula ABX$_3$, in which the position of one of the cations (at the unit cell center) is vacant. Sc atoms are located in the centers of the octahedra composed of F atoms (Figure 1). In the ScF$_3$ structure, Sc$^{3+}$ cations are located at the vertices of the primitive cubic cell. ScF$_6$ octahedra join with their vertices to form a three-dimensional framework. Volume voids in the structure give place to its looseness.

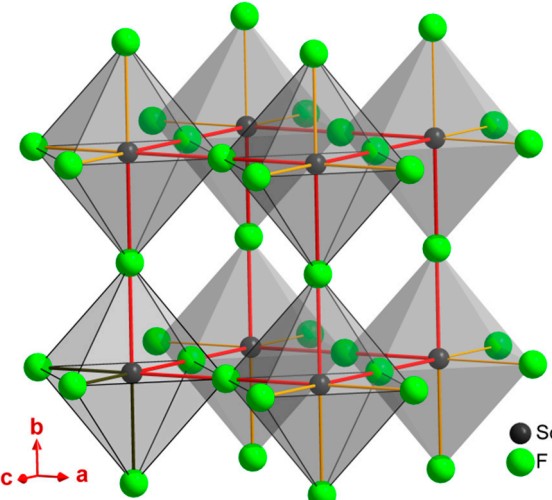

**Figure 1.** Crystal structure of cubic $ScF_3$ (space group $Pm\bar{3}m$) consisting of corner-shared regular $ScF_6$ octahedra.

$ScF_3$ exhibit a large band gap ($E_g \sim 10.5$ eV [3]) and a high chemical resistance. Among all inorganic fluorides $MF_m$ ($m$ = 1–4) $ScF_3$ has the highest melting temperature $T_m = 1822 \pm 3$ K [4]. Under normal pressure, $ScF_3$ has no solid state polymorphic transitions up to melting, but under hydrostatic pressure it transforms from $ReO_3$-type to β-$YF_3$- and $LaF_3$-types structure forms [5–8].

Structural type $ReO_3$ is very sensitive to the impurities. Cubooctahedron voids between $ScF_6$ octahedra can be filled with rather large impurity ions. Hence, physical properties determined experimentally are often distorted. Insignificant oxygen contamination in $ScF_3$ crystals leads to the structure rhombohedral distortion (sp.gr. $R32$) [8–10], and phase transition at $T = 1623 \pm 20$ K [7,8] and $1748 \pm 20$ K [10]. However, the authors [4,11] have not found any phase transformations in $ScF_3$.

$ScF_3$ exhibit a great isomorphous capacity with respect to the transition and rare-earth metals ions. Isomorphous doping enables varying of materials properties over a wide range. Solid solutions $Sc_{1-x}M_xF_3$, where $M$ = Y [12,13], Al [14], Ga and Fe [15], Ti [16] were obtained with solid-phase synthesis and then investigated. Cubic phase of $Sc_{1-x}Al_xF_3$ exists up to $x$ = 0.5 at $T$ = 1340 K, and with Al content increasing the thermal expansion coefficient of the material changes from negative to positive [14]. Phases in $Sc_{1-x}M_xF_3$ with $M$ = Ga, Fe exhibit a practically zero thermal expansion material over a wide temperature range 300−900 K [15]. The NTE of $ScF_3$ induce by the fluorine transverse oscillations normal to the Sc-F-Sc bonds and result in decreasing distance between $Sc^{3+}$ ions [17,18]. Wide-range solid solutions $Sc_{1-x}R_xF_3$ with $R$ = Yb [19,20] and Lu [21] were discovered by differential thermal analysis technique. Thus, $ScF_3$-based materials are interesting for the fundamental physical investigations of the NTE and phase-transition mechanisms.

$ScF_3$-based crystalline materials are promising for the practical applications as low-refraction thin-film coatings, matrices for doping by transition and rare-earth metals ions with small ion radius, as well as a component of materials with high fluorine-ion conductivity [22]. Study of the growth techniques and the properties of $ScF_3$ crystals can provide the possibilities to obtain thermomiotics with programmable thermal behavior, e.g., optical elements, which preserve their fixed overall dimensions over a wide temperature range.

Flux-melt method for growing $ScF_3$ with several mm dimensions, pure and doped with transition metal ions, has been described in [5,8,23–25]. LiF or NaF were used as a solvent, and the melts with content of 20–40 (mol.) % $ScF_3$ in platinum ampoules were cooled at a rate of up to 3 K/h. Synthesis $ScF_3$ nano- and microcrystals (up to 200–500 nm) from various precursors by hydrothermal synthesis was described in [26–28]. Gas–solution interface technique (GSIT) was used to obtain $ScF_3$ in the

form of nano-dimensional needles and tubes [29,30]. The above techniques enable to grow, as a rule, small single crystals.

For the first time, $ScF_3$ bulk crystals were obtained from a melt over 50 years ago by X.S. Bagdasarov with co-workers in *Shubnikov Crystallography Institute of RAS* using a direct crystallization technique [31,32]. The growth was carried out in the graphite crucibles in vacuum at the pulling rate up to 1 mm/h. The crucible was heated by means of an interaction with a non-focused electron beam, which is quite a tedious procedure. No details of the growth experiments were provided. Since then, no attempts were made to obtain $ScF_3$ crystals from the own melt. Growth of $ScF_3$ bulk crystals directly from its own melt is complicated by the high melting temperature, and NTE and high vapor pressure. Therefore, the low-temperature synthesis techniques for $ScF_3$-based crystals growth were utilized until now. Our group attempted to obtained $ScF_3$ crystal from own melt using a crucible patented by us for growing volatile substances [33]. Evaporation losses in such constriction crucible were high (more than 30 wt.%) and crystals demonstrated low optical quality. Nevertheless, we managed to fabricate a $ScF_3$ crystal elements of 0.5 $cm^3$ in size and even to measure its ionic conductivity [33].

The purpose of this paper is the development of a method for $ScF_3$ bulk crystals growth from the own melt, crystal characterization, and investigation of its some physical properties.

## 2. Materials and Methods

### 2.1. Growth Problems and Crucible Design

High melting temperature $T_m$ of $ScF_3$ and its reactivity strongly restrict the choice of the growth container materials, NTE and the high vapors pressure present almost impossible requirements to the crucible design.

Data on the $ScF_3$ saturated vapors pressure are fitted [34,35], and the vapors pressure ($P$, atm) for a wide temperature range ($T$ = 1172–1528 K) can be calculated using the following equation:

$$\lg P = (10.092 \pm 0.253) - (2.0141 \pm 0.0339) \cdot 10^4 / T.$$

These $P$ values obtained are higher than those observed during growth of $CaF_2$ or $LaF_3$ crystals by a factor of $10^2$. Extrapolation of the $P(T)$ relationship to the value $P$ = 1 atm yields the boiling temperature value $T_b$ ~ 2095 K. However, the standard accepted boiling temperature for $ScF_3$ is considered to be $T_b$ = 1880 K, i.e., only ~60 K higher than its melting temperature [36,37]. This is in contradiction with the observed practical results of the growth experiments, when the $ScF_3$ melt was strongly overheated. Hence, the boiling temperature issue is subject to a separate investigation.

Graphite is the most commonly used crucible material for a fluoride crystals growth from melt. However, $ScF_3$ melting at T ~ 1822–1875 K in the open crucible results in material evaporation losses exceeding 50 wt.% already after 1.5–2 h of the melt exposure. Generally, obtaining a pressure-tight connection in graphite details is quite a problem. There are several models of graphite crucibles [38–40] suitable for growth of crystal with high saturated vapor pressure. Sealing of the growth cell volume is achieved by using a thread connection between several graphite elements. When operating such crucibles, the melt flows into the thread connection, the result being that the crucible gets broken during its disassembly. The design suggested in [38] is essentially axially arranged and inserted into each other graphite cartridges sealed on their side surface with the condensate of the initial material during growth experiment. This model was taken as a crucible prototype used to obtaining $ScF_3$ crystals in our previous work [33]. But evaporation losses were still very high for it. For the purpose of this work, a special design graphite crucible, which enabled maintaining some excessive vapor pressure inside it, was developed [41]. Earlier, a graphite container based on this principle was successfully used by the authors to grow crystals of highly volatile fluoride compounds [42,43]. A crucible image is shown in Figure 2a.

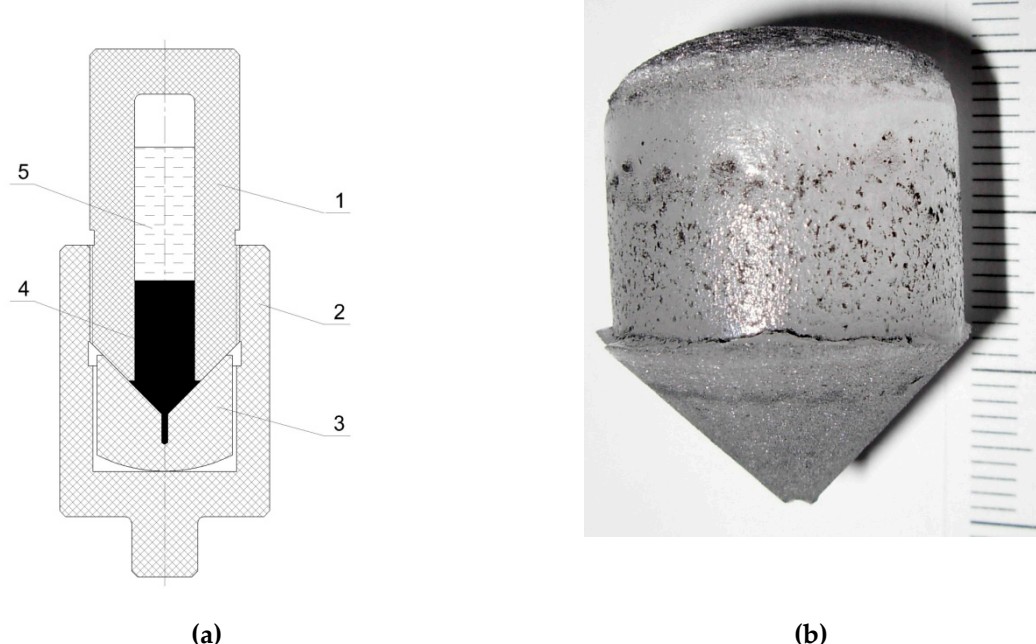

<div align="center">(<b>a</b>)         (<b>b</b>)</div>

**Figure 2.** (**a**) Schematic drawing of a self-sealing graphite crucible: 1—growth container; 2—base; 3—internal insert with the seed channel; 4—crystal; 5—melt; (**b**) As-grown $ScF_3$ crystal.

The design is essentially an upturned graphite crucible with a conical bottom which is pressed up against the base by tightening up the thread connection. The charge material placed inside provides additional sealing of the crucible bottom after melting. Utilization of such a crucible results in substantially lower material losses compared to the designed models from [38,39]. It was expected that gap availability between crucible parts will also compensate for the crystal thermal expansion during their cooling.

### 2.2. Impurity Composition

Impurity composition of crystals was determined by a spark source mass spectrometry technique based on JMS-01-BM2 mass-spectrometer (JEOL, Tokyo, Japan). The analysis results random inaccuracy was characterized by the standard deviation value 0.15–0.30. The content of inert gases and transuranic elements is below their detection limit (0.1 ppm). $ScF_3$ crystal analysis for the oxygen content was carried out with vacuum induction fusion in graphite capsules. The sensitivity of this analysis was 200 ppm.

### 2.3. Thermal Stability

Thermal stability of the $ScF_3$ crystals was investigated by heating the samples in air (under low-humidity conditions), in Pt-crucibles in a muffle furnace, in the temperature range of 500–900 K for 20–24 h. The temperature was maintained constant with an error of ±2 K.

### 2.4. X-ray Diffraction (XRD) Analysis

XRD patterns of the crystal samples, both before and after thermal treatment, were carried out on an X-ray powder diffractometer MiniFlex 600 (Rigaku, Tokyo, Japan) with $CuK_\alpha$ radiation. The diffraction peaks were recorded within the angle range 2θ from 10° to 140°. Phases were identified using the ICDD PDF-2 (2014) and Crystallography Open Databases. The unit-cell parameters were calculated by the Le Bail full-profile fitting (the Jana2006 software). A high resolution X-ray diffractometer (Rigaku SmartLab diffractometer (Tokyo, Japan) equipped with 9kW rotating anode and double-crystal Ge (220) monochromator, $MoK_{\alpha1}$ radiation) was employed to collect the X-ray rocking curve.

The samples for investigation were essentially polished plates, 0.5–5 mm thick, cut out perpendicular to the crystal growth axis.

### 2.5. Crystal Chemical Etching

Chemical etching was carried out by concentrated hydrofluoric acid as etching solution; the crystal samples were etched for 100 h at about 300 K. The surface morphology of $ScF_3$ crystals was investigated using optical microscopy and scanning electron microscopy techniques (Quanta 200 3D, FEI, Hillsboro, OR, USA).

### 2.6. Refractive Index

Refractive index $n$ and its dispersion $n(\lambda)$ for $ScF_3$ in the visible spectrum were measured using refractometric technique with the accuracy $\pm 10^{-3}$. Light sources were Hg– ($\lambda = 0.436$ and $0.546$ μm) and Na– ($\lambda = 0.589$ μm) lamps.

### 2.7. Transmission Spectra

Transmission spectra of the crystals were recorded under room temperature (RT) using Varian Cary 5000 spectrophotometer (Agilent Technologies, Santa Clara, CA, USA) in the spectral region $\lambda = 0.19$–$2.50$ μm, and AF-1 analytical Fourier-transform spectrometer (Moscow, Russia) in the region $\lambda = 2$–$16$ μm.

### 2.8. Hardness of Crystal

Crystal hardness was investigated under room temperature (RT) by the microhardness indentation technique using Wilson 2100 Tukon Vickers/Knoop hardness tester (Aachen, Germany) operating in automatic mode.

The Vickers hardness ($H_V$) values at different loads were calculated using the following equation:

$$H_V = 1.854 P/d^2 \ [\text{kgf/mm}^2],$$

where $P$ is the applied load and $d$ is the diagonal length of the indentation imprint (1 kgf = 9.807 N). Measurement error was not in excess of 3%.

### 2.9. The Thermal Conductivity

The thermal conductivity $k(T)$ of the crystal was measured by an absolute steady-state axial heat flow technique in the temperature range of 50–300 K. The measurement procedure was described in detail elsewhere [44]. The sample was a non-oriented parallelepiped $6 \times 6 \times 20$ mm$^3$ in size. The error in determining the absolute $k$ value did not exceed $\pm 5\%$.

### 2.10. The Electrical Conductivity

*The electrical conductivity* σ of the crystals was determined by impedance spectroscopy. The measurements were carried out in the frequency range of $10^{-2}$–$10^7$ Hz in the temperature range of 350–830 K using a Novoterm-HT 1200 system with Alpha-A+ZG4 (Novocontrol Technologies GmbH & Co KG, Montabaur, Germany) impedance tester. Silver paste (Leitsilber) as current-conducting electrodes was used. The relative measurement error was less than 1%.

## 3. Results and Discussion

### 3.1. Crystal Growth

Growth of $ScF_3$ crystals was carried out by the Bridgman technique in a double-zone growth chamber with resistive heating in the high pure He atmosphere, in the graphite heat unit.

$ScF_3$ charge material was obtained in a laboratory from $Sc_2O_3$ oxide (99.95%, Novosibirsk rare metals plant) by transferring it into chloride solution. Then it was deposited with concentrated HF acid followed by drying, vacuum calcination and melting in the atmosphere of polytetrafluoroethylene pyrolysis products. The growth chamber was evacuated to provide residual pressure ~$10^{-3}$ Pa using a turbomolecular pump system. Mixture of $NH_4HF_2$ (97.5% Sigma-Aldrich) and $CF_4$ (99.999%) were used as fluorinating agents during growth. Temperature gradient in the growth zone was ~80 K/cm, crucible pulling rate - 5 mm/h, cooling rate of grown crystals ~100 K/h.

$ScF_3$ single crystals of optical quality have been grown in the self-sealing graphite crucibles described above. The crystals were colorless, transparent; losses of the substance during crystallization process were 4 wt.% only, no cracks and light-scattering centers were observed. Crystal $ScF_3$ boules 25 mm in diameter and 50 mm long were obtained. One of the as-grown $ScF_3$ crystals is shown in Figure 2b. With the spontaneous crystal seeding, the $ScF_3$ growth axis was close to [111] crystallographic direction.

Impurities content is provided in Table 1 showing that the major contribution to the chemical composition is made by cations of alkali-earth and rare-earth metals resulting from the original $Sc_2O_3$, trace impurity of chlorine resulting from the $ScF_3$ reagent synthesis, and oxygen.

**Table 1.** Results of chemical analysis of $ScF_3$ crystals.

| Element | Content, ppm | Element | Content, ppm | Element | Content, ppm |
|---------|-------------|---------|-------------|---------|-------------|
| Li | 0.08 | Ga | <0.02 | Pr | **20** |
| Be | <0.01 | Ge | <0.02 | Nd | **40** |
| B | 0.04 | As | <0.02 | Sm | **8** |
| O | **300** | Se | <0.02 | Eu | **1** |
| F | Basis | Br | 0.6 | Gd | **8** |
| Na | 0.2 | Rb | <0.02 | Tb | **1** |
| Mg | **6** | Sr | **4** | Dy | **15** |
| Al | 0.2 | Y | 260 | Ho | **4** |
| Si | 1 | Zr | <0.05 | Er | **15** |
| P | <0.02 | Mo | <0.05 | Tm | **3** |
| S | 2 | Ru | <0.05 | Yb | **40** |
| Cl | **5** | Rh | <0.05 | Lu | **6** |
| K | 0.5 | Pd | <0.05 | Hf | <0.1 |
| Ca | **580** | Ag | <0.05 | W | <0.1 |
| Sc | Basis | Cd | <0.05 | Re | <0.1 |
| Ti | <0.01 | In | <0.05 | Os | <0.2 |
| V | <0.01 | Sn | <0.05 | Ir | <0.1 |
| Cr | <0.01 | Sb | <0.05 | Pt | <0.2 |
| Mn | <0.01 | Te | <0.05 | Au | <0.2 |
| Fe | 2 | I | 0.4 | Hg | <0.2 |
| Co | <0.02 | Cs | <0.1 | Tl | <0.1 |
| Ni | <0.02 | Ba | **9** | Pb | <0.2 |
| Cu | <0.02 | La | **40** | Bi | <0.2 |
| Zn | <0.02 | Ce | **4** | U | <0.2 |

### 3.2. Crystal Structure and Quality

The assignment of crystals to the structure type $ReO_3$ ($Pm\bar{3}m$ space group) has been confirmed by XRD. XDR diffraction patterns of the as-grown and heat-treated $ScF_3$ crystals are shown in Figure 3a. The cubic lattice parameter of the as-grown $ScF_3$ crystal is $a = 4.01401(3)$ Å at 295 K, which confirms well with the published data (coincides with the standard patterns PDF #79-8099, COD ID 4127282). Crystal density $\rho = 2.601(3)$ g/cm$^3$ (measured by hydrostatic weighing in distilled water) is insignificant lower than the theoretical density value.

Thermal treatment of crystals in the air showed that $ScF_3$ exhibit stability and do not loose transparency when heated up to $T = 773$ K. With increasing the temperature, the crystals surface was

covered by additional phase $Sc_2O_3$ ($Ia\overline{3}$ space group, standard patterns PDF #84-1880, COD ID 4326666) with the lattice parameter $a = 9.8420(3)$ Å (see Figure 3a), the phase being due to pyrohydrolysis of $ScF_3$. According to [1,37], the reaction of $ScF_3$ pyrohydrolysis follows the equation:

$$2ScF_3 + 3H_2O = Sc_2O_3 + 6HF,$$

without formation of intermediate oxofluoride phases specific for other $RF_3$.

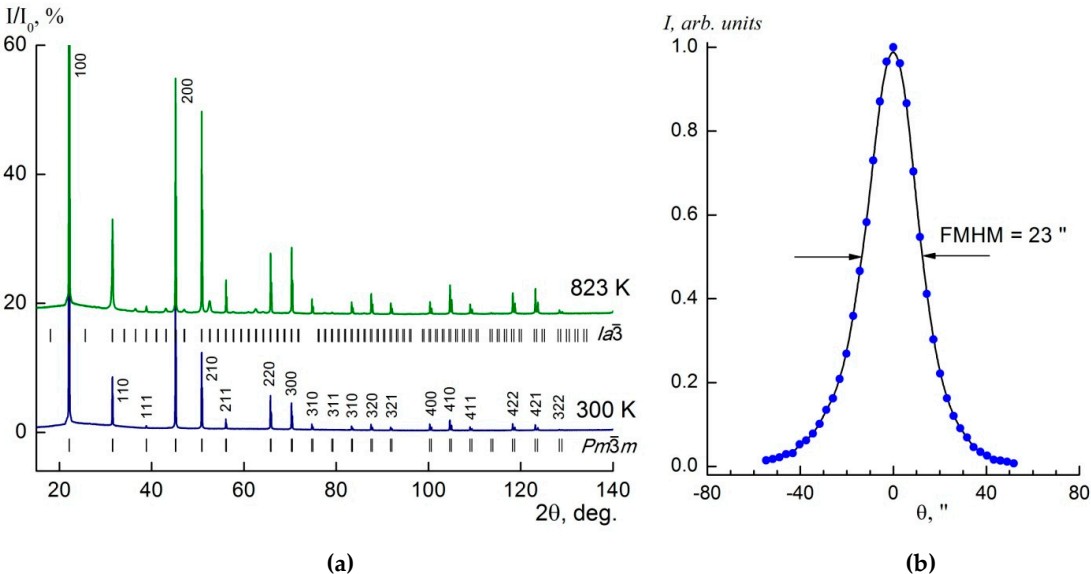

**Figure 3.** (**a**) XDR patterns of the as-grown and heat-treated at $T = 823$ K ScF3 crystals. The Bragg reflections positions for the $Pm\overline{3}m$ and $Ia\overline{3}$ space groups are indicated; (**b**) X-ray rocking curve of $ScF_3$ crystal 222 reflection.

The X-ray double-crystal rocking curve of the 222 reflection is shown in Figure 3b. The full width at half maximum (FWHM) intensity is 23" (0.0064°). The profile of the rocking curve is fairly smooth, symmetric, without splitting due to twinning. From this point of view, the measured curve indicates a relatively high quality of the $ScF_3$ crystal. The peak broadening (the typical FWHM for perfect semiconductor single crystals is about 5-10") is most likely due to the presence of crystal lattice deformations (strains) in the samples.

Selective chemical etching is an effective and simple method for investigation of the crystal surface morphology and structure imperfections. High chemical stability of $ScF_3$ hinders the search for an effective selective etchant. In our work HF was choose as an etchant during the long period of interaction. Wide mosaic structure and grain boundaries aggregations appears in the noncentral crystal parts (Figure 4a) due to NTE and arising strong strain in places where the crystal is in contact with the crucible walls. The triangle shape of dislocation etching pits is show in Figure 4b. Etching pits size varied in the range of 0.1–0.5 μm², and their density was ~1*10⁷ cm⁻² in the central part but it increased up to ~6*10⁷ cm⁻² in the peripheral crystal area. Therefore, for improving the $ScF_3$ crystals quality (removing lattice strains and decreasing density of dislocations) it is necessary to post-growth $ScF_3$ crystal annealing.

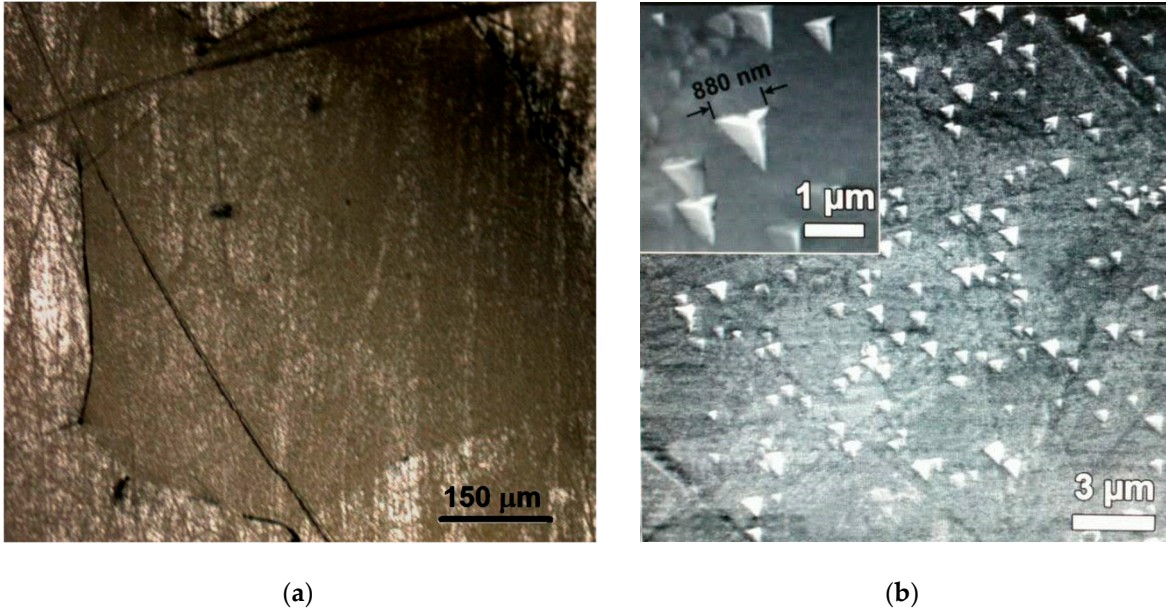

(**a**)                                                      (**b**)

**Figure 4.** (**a**) The blocky structure of peripheral crystal areas; (**b**) Dislocation etching pits pattern on (111) crystallographic $ScF_3$ crystal face.

### 3.3. Optical Properties

The RT transmission spectrum of the $ScF_3$ crystal over the wide range is shown in Figure 5a. The observed absorption bands and low transparency in UV spectral regions are attributed with presents of uncontrollable impurities, primarily, oxygen-containing. Transmission cutoff in IR-region is located at 8.7 μm. Additional deep purification of initial charge can significantly improve the transmission of $ScF_3$ crystals in the short-wavelength part of the transparency window. The last one is being very sensitive to the crystal impurity contaminations. A wide band gap of $ScF_3$ crystals makes it a promising candidate as a new transparent matrix in vacuum UV spectral range comparable to such material as $LaF_3$ et al. [45].

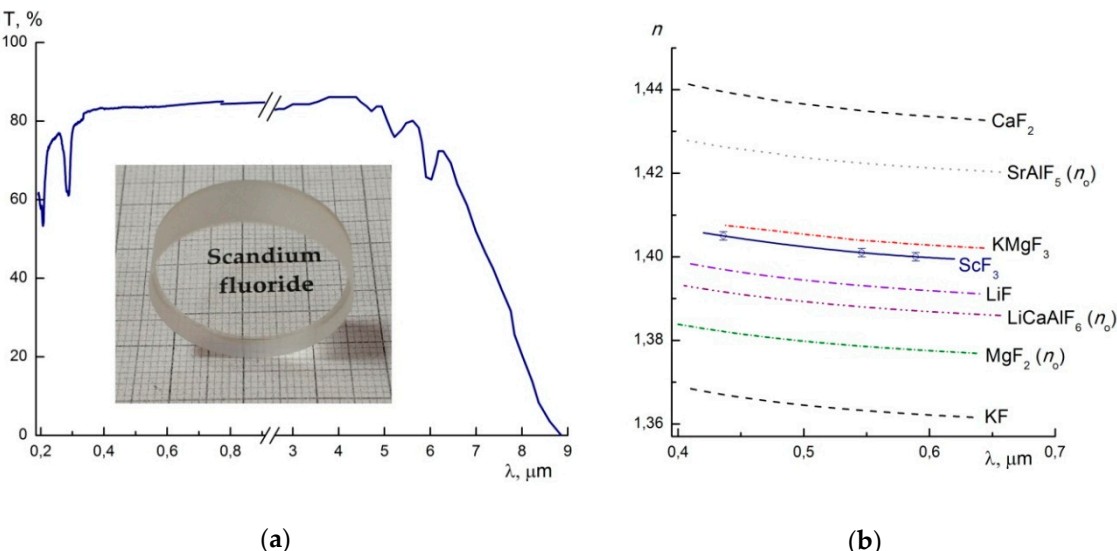

(**a**)                                                      (**b**)

**Figure 5.** (**a**) Transmission spectrum of 3 mm thick crystal and fabricated $ScF_3$ optical element; (**b**) Refractive index dispersions of some fluoride crystals.

For the crystalline $ScF_3$, the refractive index was measured $n_D = 1.400(1)$ and its dispersion dependence was obtained in the visible region (see Figure 5b). Dependence $n(\lambda)$ for some low-refractive fluoride crystals [46–48] are also provided in Figure 5b for the comparison purposes.

Mathematical processing of dispersion dependence $n(\lambda)$ of $ScF_3$ crystals was carried out using one-term Sellmeier equation:

$$n^2(\lambda) = 1 + \frac{A\lambda^2}{\lambda^2 - \lambda_0^2},$$

where $A = 0.9435(1)$ is the multiple related to the oscillators number and strength, $\lambda_0 = 0.794(1)$ μm is the characteristic wavelength.

$ScF_3$ crystals refer to the low-refractive and low-dispersion materials. These materials can be considered as a promising isotropic material for antireflection coatings, on a par with the isostructural fluoroperovskite $KMgF_3$ crystal. Such materials with lower refractive index as KF and LiF are hygroscopic, and $MgF_2$ and $LiCaAlF_6$ – birefringent. It is worth noting that the data on the refractive indices for $ScF_3$ thin-film [49] are substantially higher than those for bulk crystals and coincides with vitreous silica.

### 3.4. Thermal Conductivity Measurements

Measurement results for thermal conductivity $k(T)$ of $ScF_3$ crystal are shown in the graphical form in Figure 6a. Experimental curves of $CaF_2$ (space group $Fm\bar{3}m$, Z = 4) [50] and $LaF_3$ (space group $P\bar{3}c1$, Z = 6) [51] with different from $ScF_3$ crystals structure are also shown for the comparison. As can be seen in Figure 6a, absolute values for $ScF_3$ thermal conductivity within the observed temperature range are intermediate between the relative values for $CaF_2$ and $LaF_3$. However, temperature dependence $k(T)$ for $ScF_3$ is noted for its weakness. Within the temperature range, the value of $k$ changes only by a factor of 2, i.e., from 20.5 $Wm^{-1}K^{-1}$ at $T = 50$ K to 9.6 $Wm^{-1}K^{-1}$ at $T = 300$ K. The nature of the temperature dependence is slightly different from that determined by logarithmic function of the following type:

$$k(T) = -6.171 \times ln\ T + 44.4\ [Wm^{-1}K^{-1}]$$

The extrapolation of our experimental data into the high temperature range ($T$ = 400–1500 K) provides a rather accurate coincidence with the calculated ab initio values $k(T)$ obtained by authors [52].

The availability of experimental data on the $ScF_3$ heat capacity [13] makes it possible to calculate temperature dependence of the mean free path $l(T)$ of the phonons in this crystal. For the calculation, use has been made of the known Debye expression:

$$k = C\nu\ l/3,$$

where $C$ is the heat capacity of a crystal volume unit, $\nu$ is the mean phonons velocity. Thermal phonons mean velocity $\nu$ was estimated as follows. Using elastic constants $c_{11} = 233$, $c_{12} = 17$ and $c_{44} = 18$ GPa [53,54] and referring to the known relationships for the cubic crystal system, calculations were made for the longitudinal wave velocity $\nu_l = 7.48$ km/s and velocities of two transverse waves – $\nu_{S_1} = 2.67$ km/s and $\nu_{S_2} = 6.48$ km/s. The averaging was made as per the known formula:

$$\frac{3}{\nu^3} = \frac{1}{\nu_1^3} + \frac{1}{\nu_{S_1}^3} + \frac{1}{\nu_{S_2}^3}.$$

As a result, $\nu = 3.71$ km/s for $ScF_3$ crystals. The results of calculation of the phonons mean free path as a function of temperature $l(T)$ are shown in Figure 6b. The similar data for $CaF_2$ [50] and $LaF_3$ [51] crystals are also provided here for the comparison.

One can note the weak relative dependence $l(T)$ obtained for $ScF_3$, even for lowest temperatures. Such temperature dependence is specific for materials with disturbances in the periodicity of their crystal field [55]. Such disturbances perhaps are related to the $ScF_3$ structure intrinsic imperfection.

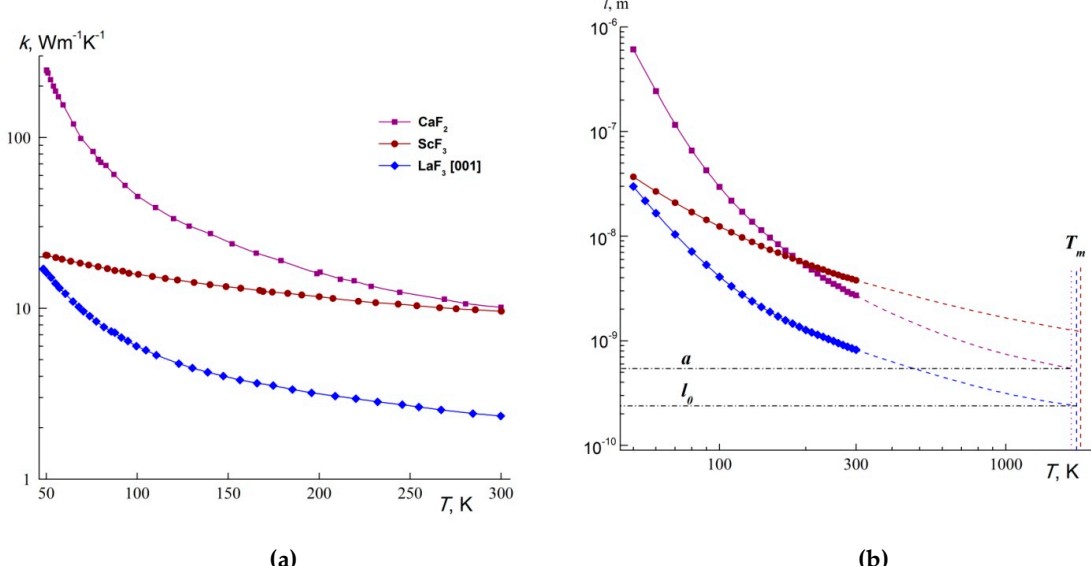

**Figure 6.** (**a**) Temperature dependences of the thermal conductivity $k(T)$ (**a**) and the phonon mean free path $l(T)$ (**b**) of $CaF_2$, $ScF_3$ and $LaF_3$ crystals.

Indeed, the investigation of $ScF_3$ crystals using the electron paramagnetic resonance technique [24] showed that there are lines in the spectrum that can be referred to as the local defects in the crystal structure. $Sc^{3+}$ vacancies in the octahedral cation positions (Frenkel defect i.e., the vacancies being, perhaps, related to the fact that $Sc^{3+}$ cations occupy other initially vacant positions with a large CN) were considered as the defects. The evident signs of local distortion in the cubic symmetry crystal field are also indicated. (see also [25]). The electrical conductivity of undoped $ScF_3$ crystals with application of the anti-Frenkel anion defects model ($F^-$ ions transition into cubooctahedron void and vacancies formation in the main anion sublattice) and their associates was considered in Reference [56].

It does not seem possible to divide contributions of the above defects into the thermal resistance of the $ScF_3$ crystal. Taking account of the substantial role of $Sc^{3+}$ cations in maintaining crystal structure stability one may assume a more substantial effect in thermal phonons dispersion on the cation defects (without referring to the number of these defects). Relationship between fluorine ion conductivity and the anion sublattice disorder in fluoride crystals, and observed correlation with thermal conductivity was reported in Reference [57].

Let us note the following distinction of the obtained results. Extrapolation of $l(T)$ into the melting temperature value $T_m$ (Figure 6b) provides the value $l_{min} \approx a$ for both $CaF_2$ crystal and other crystal with fluorite structure [58], whereas for $LaF_3$ $l_{min}$ is close to the mean interstitial distance $l_0$. In case of $ScF_3$ crystals the value $l_{min}$ is three-fold the parameter $a$ of the unit cell. Perhaps, specifics in the behavior of $l(T)$ and $k(T)$ are related with the special type of individual modes anharmonicity of $ScF_3$ crystal thermal vibrations. Necessity of description of $ScF_3$ phonon spectrum with the application of a quantum quartic oscillator model was noted in Reference [17].

### 3.5. Mechanical Properties

Dependence of $ScF_3$ microhardness on indentation load is shown in Figure 7. Vickers microhardness, under the load $P = 0.3$ H, is $H_V = 2.58(4)$ GPa which is higher than that of both alkali-earth ($CaF_2$, $SrF_2$ [59]) and rare-earth ($LaF_3$ [60]) fluoride crystals. When the load was $P > 0.4$ N the area around indentation exhibited cracks in the material. $ScF_3$ crystal refers to the group of solid materials with a high tendency to brittle failure.

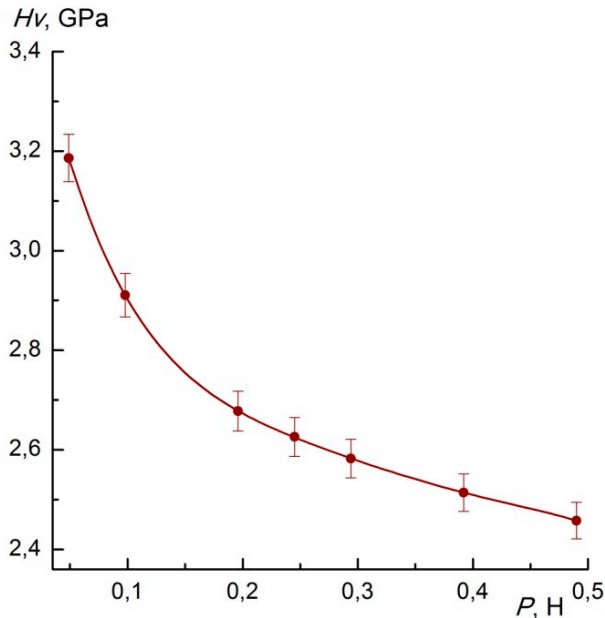

**Figure 7.** Vickers microhardness of ScF$_3$ crystals as a function of the load applied on the (111) plane.

### 3.6. Ionic Conductivity

Temperature dependences of the ionic conductivity of the ScF$_3$ and some rare earth fluoride crystals are shown in Figure 8. The anionic transport parameters for ScF$_3$ are well described by the Arrhenius-Frenkel equation:

$$\sigma T = A\exp(-\Delta H_\sigma/kT),$$

where $A = 9.7 \times 10^3$ SKcm$^{-1}$ is the pre-exponential factor for electrical conductivity, $\Delta H_\sigma = 1.13 \pm 0.05$ eV is the ion transport activation enthalpy, $k$ is the Boltzmann's constant, $T$ is the temperature.

Electrical conductivity of ScF$_3$ crystals changes by the order of 8 within the presented temperature range 350–833 K. Ionic conductivity is $\sigma = 2 \times 10^{-9}$ Sm/cm at $T = 573$ K. These results coincide with data previously published in Reference [33] within experimental error. Such low conductivity of the ScF$_3$ single crystal confirms data for the polycrystalline ScF$_3$ solid [56]. The mobility of highly charged Sc$^{3+}$ cations is unlikely; the ion transport in ScF$_3$ crystals is related to anti-Frenkel defects in the anion (fluorine) crystal sublattice that is created due to the F$^-$ ions transition into interstitial positions and simultaneous formation of anion vacancies [33]. The same crystal-chemical specifics belong to CaF$_2$ crystals featuring low intrinsic conductivity [61,62]. As can clearly be seen in Figure 8, ScF$_3$ crystal is substantially inferior to another $R$F$_3$, such as HoF$_3$ [63], recently studied similar TbF$_3$ [64] (with β-YF$_3$-type) and tysonite LaF$_3$ crystals [61,65,66] in terms of the ionic conductivity.

Low conductivity of ScF$_3$ crystal (compared to other $R$F$_3$ crystals) is caused by the Sc$^{3+}$ ions low coordinating ability (CN = 6), which is not typical for rare-earth cations in fluorides, their low electron polarizability ($\alpha_{cat} = 1.1$ Å$^3$ [56]), and by strong Sc−F couplings in crystal structure (ScF$_3$ has the highest melting temperature among all fluorides).

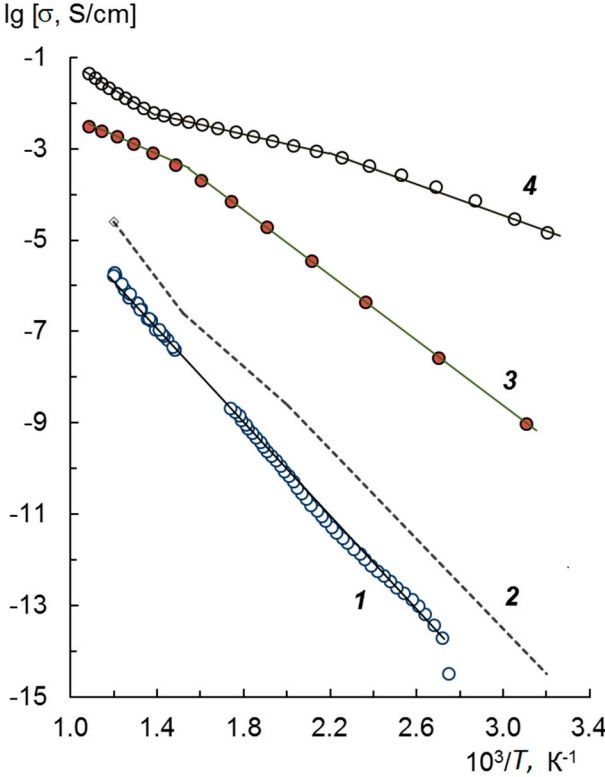

**Figure 8.** Comparative data on the ionic conductivity temperature dependencies $\sigma(T)$ of the ScF$_3$ single crystal [this work and [33]] (*1*), polycrystalline ScF$_3$ [56] (*2*), HoF$_3$ [63] (*3*) and LaF$_3$ [65] (*4*) single crystals.

## 4. Summary

A novel bulk ScF$_3$ crystal growth procedure with a self-sealing graphite crucible was designed. Optical quality ScF$_3$ crystals were grown successfully from the own melt by the Bridgman technique. It was noted that the density of the etching pits is very high (more than $10^7$ cm$^{-2}$), which will require further development of the post-growth annealing procedure to reduce it.

Grown cubic ScF$_3$ crystals are superior to such materials as LaF$_3$ and CaF$_2$ due to their unique mechanical and optical properties. Crystals are transparent in broad spectral range up to 8.7 µm and save optical transparency under heating on air up to $T = 773$ K; observed weak absorption bands are related to the oxygen contamination and a more chemically pure initial charge should be used to solve this problem. ScF$_3$ crystals shows the isotropy of optical properties and low refractive index $n_D = 1.400(1)$, which is close to the data for other cubic LiF and KMgF$_3$ crystals.

The ScF$_3$ thermal conductivity examination demonstrate $k = 9.6$ Wm$^{-1}$K$^{-1}$ at room temperature, which coincides with CaF$_2$ crystals. The Vickers hardness of the ScF$_3$ crystals strongly depends on the applied load and average is $H_V \sim 2.6$ GPa, which is substantially more than similar data for LaF$_3$ and CaF$_2$ crystal. Among others studied $R$F$_3$, these crystals show a very low ionic conductivity ($\sigma = 4 \times 10^{-8}$ Scm$^{-1}$ at 673 K) cause to structural defects in the fluorine sublattice.

We believe that our data on crystalline ScF$_3$ are important for the rare earth materials science and demonstrate the competitive possibilities for its application in photonics and optical instrumentation.

**Author Contributions:** D.K., I.B., N.A., P.P. performed the experiments, prepared figures and manuscript; D.K., I.B. performed single crystal experiments; N.A. and P.P. analyzed crystal quality; V.G. and N.S. performed conductivity investigation; T.G. performed optical investigation; P.P. performed thermal conductivity investigation; D.K., I.B., P.P. analyzed the data, interpreted experiments; D.K. and I.B. provided the idea, designed the experiments; D.K. coordinated the group.

**Funding:** This work was supported by the Russian Foundation for Basic Research (projects nos. 17-00-00118, 19-02-00877) in the part concerning growth of crystals and by the Ministry of Higher Education and Science of the Russian Federation within the State assignments of the Federal Scientific Research Centre "Crystallography and Photonics" of the Russian Academy of Sciences and Petrovsky Bryansk State University (nos. 3.8326.2017/8.9) in the part concerning investigation and analysis of crystal characteristics.

**Conflicts of Interest:** The authors declare no conflict of interest.

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
