# Peer review of "Growth from the Melt and Properties Investigation of ScF3 Single Crystals"

_crystals, doi:10.3390/cryst9070371_

Round 1

Reviewer 1 Report

Authors described detailly crystal growth conditions, but I could not find in the paper a comparison between different crystal growth methods and prevalence of the proposed one. Please complete the deficiency in chapter 2.1. I can only guess that it is also the geometrical size. But how to compare quality (rocking curves) between the methods.

The Summary does not contain conclusions resulting from performed investigations. Please complete the deficiency.

When authors decide to present Table 1, they should also discuss the defect structure of the crystal. Some of them one can see in Fig. 5. It should be discussed bearing in mind a kind of defects.

Author Response

Comment 1

Authors  described detailly crystal growth conditions, but I could not find in the  paper a comparison between different crystal growth methods and prevalence of  the proposed one. Please complete the deficiency in chapter 2.1. I can only  guess that it is also the geometrical size. But how to compare quality  (rocking curves) between the methods.

Reply 1

Thank you for  your comments. Of course, it is advisable to add a comparison between  different growth methods. But, there is small information about the methods of  growth bulk ScF3 and the details of the growth process are not described, the  quality of crystals has not been evaluated at all (only in terms of  transparent, etc.), then we mainly compare the size of crystals. We have  entered data on the geometric dimensions of crystals obtained by various  methods, as well as information on rocking curves. It is not possible to  compare rocking curves, as data from other authors are not available. The  curve obtained by us will be successfully used for a comparative analysis after  improving the quality of the crystals after annealing (similarly, for example,  as described in the article https://doi.org/10.1016/S0925-3467(03)00115-0).

Comment 2

The Summary does  not contain conclusions resulting from performed investigations. Please  complete the deficiency.

Reply 2

We have radically revised section 4 and Summary.  See manuscript main text

Comment 3

When authors  decide to present Table 1, they should also discuss the defect structure of  the crystal. Some of them one can see in Fig. 5. It should be discussed  bearing in mind a kind of defects.

Reply 3

We took into account the comments and  added a discussion of the data in Table 1. Relationship of impurities with the  defect structure of the ScF3 crystal  cannot be identified. We are simply stating the observed fact. Direct  correlation can be observed only for the transmission spectra and the oxygen  contamination in the crystals.

Reviewer 2 Report

Dear Authors,

Your work is perfectly suited for publication in Crystals. I would only have two points in order to ameliorate:

- You point the PDF file for ScF6. Actually, the structure itself is not defined in such a file (the atomic positions are not in, only a list of lines with cell parameters). You would prefer the Crystallography Open Database entries at http://www.crystallography.net/cod/index.php, please add the corresponding codes available for your phases

- Similarly, you should upload your physical properties at the Materials Properties Open Database at  http://mpod.cimav.edu.mx, these will be very valuable to the community.

You will find on these sites the original articles to cite too.

Sincerely

Author Response

Comment 1

You point the  PDF file for ScF6. Actually, the structure itself is not defined in such a  file (the atomic positions are not in, only a list of lines with cell  parameters). You would prefer the Crystallography Open Database entries at  http://www.crystallography.net/cod/index.php, please add the corresponding  codes available for your phases.

Reply 1

Thank you for appreciation of our article  and your comments. Of course, you are right. We have entered the appropriate  COD ID’s for these compounds.

Comment 2

Similarly, you  should upload your physical properties at the Materials Properties Open  Database at  http://mpod.cimav.edu.mx,  these will be very valuable to the community.

Reply 2

Thank you for this note. We will  certainly do it after publishing this article.

Reviewer 3 Report

The authors state that they first grew ScF3 single crystals from the melt by the Bridgman technique using a specially designed graphite crucible [40]. They studied a complex of physical properties, including electrical conductivity in a range of 350–650 K. However, the authors are silent about the fact that similar results were published in the journal Crystallography Reports. 2016. V. 61. No. 2. P. 270–274. The authors do not refer to this article. In the previously published paper, ScF3 single crystals were also grown from the melt by the Bridgman technique using a specially designed graphite crucible; electrical conductivity was also measured for the first time in the interval 363–832 K. Fig. 9 in submitted article is a part of Fig. 2 in the previously published article. It looks like plagiarism. How can authors explain this situation? The other results do not cause questions. The article should be redone taking into account the previously published work.

Article is written carelessly. ReO3: what is Re? Rhenium or rare earth? There is also the abbreviation REE in the text. A single designation should be used for the rare earth element. There are a lot of misprints in the text.

According to printed symbols for crystallographic items (International Tables for Crystallography (2006). Vol. A, Chapter 1.1, pp. 2–3.), hkl are indices of the Bragg reflection (Laue indices) from the set of parallel equidistant net planes (hkl); (hkl) are indices of a crystal face, or of a single net plane (Miller indices). So, reflections on Fig. 3 and in the text should be printed without brackets.

Who is the V.V. who performed the conductivity studies?

Authors should improve English, especially pay attention to punctuation.

Round 2

Reviewer 3 Report

I read the revised version of the article. Now everything is all right. I'm satisfied. The article may be accepted for publication. Moderate English changes required.